# α-actinin accounts for the bioactivity of actin preparations in inducing STAT target genes in *Drosophila melanogaster*

Oliver Gordon[1†], Conor M Henry[1†], Naren Srinivasan[1‡], Susan Ahrens[1§], Anna Franz[2], Safia Deddouche[1#], Probir Chakravarty[3], David Phillips[4¶], Roger George[5], Svend Kjaer[5], David Frith[6], Ambrosius P Snijders[6], Rita S Valente[7], Carolina J Simoes da Silva[8], Luis Teixeira[7], Barry Thompson[9], Marc S Dionne[10], Will Wood[11], Caetano Reis e Sousa[1*]

[1]Immunobiology Laboratory, The Francis Crick Institute, London, United Kingdom; [2]Department of Biochemistry, Biomedical Sciences, University of Bristol, Bristol, United Kingdom; [3]Bioinformatics, The Francis Crick Institute, London, United Kingdom; [4]Genomics-Equipment Park, The Francis Crick Institute, London, United Kingdom; [5]Structural Biology, The Francis Crick Institute, London, United Kingdom; [6]Proteomics, The Francis Crick Institute, London, United Kingdom; [7]Instituto Gulbenkian de Ciencia, Oeiras, Portugal; [8]Department of Life Sciences, Imperial College London, London, United Kingdom; [9]Epithelial Biology Laboratory, The Francis Crick Institute, London, United Kingdom; [10]MRC Centre for Molecular Bacteriology and Infection, Imperial College London, London, United Kingdom; [11]Edinburgh Medical School, MRC Centre for Inflammation Research, University of Edinburgh, Edinburgh, United Kingdom

**\*For correspondence:**
Caetano@crick.ac.uk

[†]These authors contributed equally to this work

**Present address:**
[‡]GlaxoSmithKline, Stevenage, United Kingdom; [§]Voisin Consulting Life Sciences, Surrey, United Kingdom; [#]Open Innovation Access Platform, Strasbourg, France; [¶]Thermo Fisher Scientific, Renfrew, United Kingdom

**Competing interests:** The authors declare that no competing interests exist.

**Abstract** Damage-associated molecular patterns (DAMPs) are molecules exposed or released by dead cells that trigger or modulate immunity and tissue repair. In vertebrates, the cytoskeletal component F-actin is a DAMP specifically recognised by DNGR-1, an innate immune receptor. Previously we suggested that actin is also a DAMP in *Drosophila melanogaster* by inducing STAT-dependent genes (*Srinivasan et al., 2016*). Here, we revise that conclusion and report that α-actinin is far more potent than actin at inducing the same STAT response and can be found in trace amounts in actin preparations. Recombinant expression of actin or α-actinin in bacteria demonstrated that only α-actinin could drive the expression of STAT target genes in *Drosophila*. The response to injected α-actinin required the same signalling cascade that we had identified in our previous work using actin preparations. Taken together, these data indicate that α-actinin rather than actin drives STAT activation when injected into *Drosophila*.
DOI: https://doi.org/10.7554/eLife.38636.001

## Introduction

Metazoan organisms need to be able to recognise damaged tissues in order to induce processes that promote tissue repair and that prevent infection caused by barrier breach. Tissue damage is generally accompanied by cell death and release of damage-associated molecular patterns (DAMPs), molecules that are sequestered within healthy cells but become exposed to the extracellular milieu upon loss of membrane integrity and that trigger inflammation, modulate immunity or promote tissue repair (*Rock et al., 2010*; *Zelenay and Reis e Sousa, 2013*). Examples of DAMPs include ATP, uric acid, IL-33, IL1α, RNA and DNA (*Zitvogel et al., 2010*), as well as actin, a ubiquitous and

abundant component of the cytoskeleton of all eukaryotes. Exposure of filamentous (F-) actin by dead cells can be recognised by an innate immune receptor known as DNGR-1 (aka CLEC9A) expressed by specialised leucocytes (*Ahrens et al., 2012*; *Hanč et al., 2015*; *Zhang et al., 2012*). DNGR-1 is only found in mammals but we recently provided evidence that cytoskeletal exposure may be sign of cell damage also in *Drosophila melanogaster*. In *Drosophila*, mechanical and other stresses elicit JAK-STAT pathway signalling, which is implicated in stem cell mobilisation and tissue repair (*Ekengren and Hultmark, 2001*; *Ekengren et al., 2001*; *Agaisse et al., 2003*; *Brun et al., 2006*; *Jiang et al., 2009*). We found that injection of purified actin into adult flies led to selective induction of STAT target genes in the fat body, the *Drosophila* equivalent of the mammalian liver (*Srinivasan et al., 2016*). The response to actin required the NADPH oxidase Nox, the Src family kinase Src42A, and the adapter Shark and led to an autocrine and paracrine amplification loop that involved the cytokine Upd3 and the JAK-STAT-coupled cytokine receptor Domeless (*Srinivasan et al., 2016*). Here, we revise our interpretation of those data and report that alpha-actinin (α-actinin), a cytoskeletal protein tightly associated with F-actin, is a more potent inducer of STAT target genes than actin. Like the response to actin, the response to α-actinin requires Nox, Src42A and Shark. Notably, α-actinin can be found in trace amounts in the purified actin preparations that we had used in our initial study and recombinant actin expressed in bacteria and devoid of α-actinin is no longer capable of eliciting the STAT response upon injection into flies. We conclude that α-actinin rather than actin is the key trigger of STAT activation upon injection into *Drosophila*, suggesting that distinct cytoskeletal proteins can serve as DAMPs across species.

## Results and discussion

In our previous study, actin purified from human platelets or rabbit muscle or recombinant actin purified from insect cells elicited the expression of STAT-responsive genes in the fat body when injected into adult *Drosophila melanogaster* (*Srinivasan et al., 2016*). While assessing the ability of other cytoskeletal proteins to elicit a similar response, we found that myosin, α-actinin, and, to a lesser degree, tubulin could also drive induction of the STAT responsive gene *TotM* (*Figure 1a*). On a per molecule basis, α-actinin was the most potent trigger and was superior to myosin, the second most potent inducer (*Figure 1a,b*). Robust induction of *TotM* was observed as early as 6 hr post α-actinin injection and was sustained above control levels for at least two days (*Figure 1c*). The ability of α-actinin to induce STAT-responsive gene induction was independently reproduced in three laboratories (C.R.S, M.D., L.T.), underscoring the robustness of the result (data not shown). Like actin itself, α-actinin is a component of the cytoskeleton in all higher eukaryotes, where it crosslinks and stabilises actin filaments (*Ribeiro et al., 2014*). Given its association with actin, α-actinin could therefore be present as a contaminant in purified actin preparations, including the ones used in our studies. Consistent with that possibility, mass spectrometry analysis revealed that α-actinin is the major contaminant of purified actin preparations and accounts for approximately 0.4% of total protein (*Figure 1d*). Western blot analysis revealed the presence of immunodetectable α-actinin in actin preparations, confirming the mass spectrometry results (*Figure 1e*). Notably, dose response curves showed that α-actinin was >100 fold more potent than actin on a per molecule basis at eliciting the *Drosophila* STAT response (*Figure 1f*). Therefore, it is possible that contamination with α-actinin accounts for the activity of injected actin preparations in *Drosophila*.

To address this possibility, we tried to deplete α-actinin from actin preparations. However, we failed to satisfactorily separate actin and α-actinin using multiple approaches, including size exclusion chromatography, ionic strength chromatography or immunodepletion with 14 different antibodies (data not shown). We therefore pursued an alternative strategy of testing recombinant actin and α-actinin expressed in BL21 *E. coli*, an organism that does not have an actin-based cytoskeleton and in which, therefore, co-purification of α-actinin with actin is not possible. We first assessed the functional integrity of the recombinant proteins purified from bacteria. Using a pelleting assay (*Ahrens et al., 2012*), we confirmed that recombinant actin was able to form filaments when incubated in polymerisation-favouring conditions (*Figure 2a*) and that α-actinin was able to associate with those filaments (*Figure 2b*). Having ascertained that actin and α-actinin are functional and, therefore, correctly folded, we next tested whether they could elicit the expression of STAT-

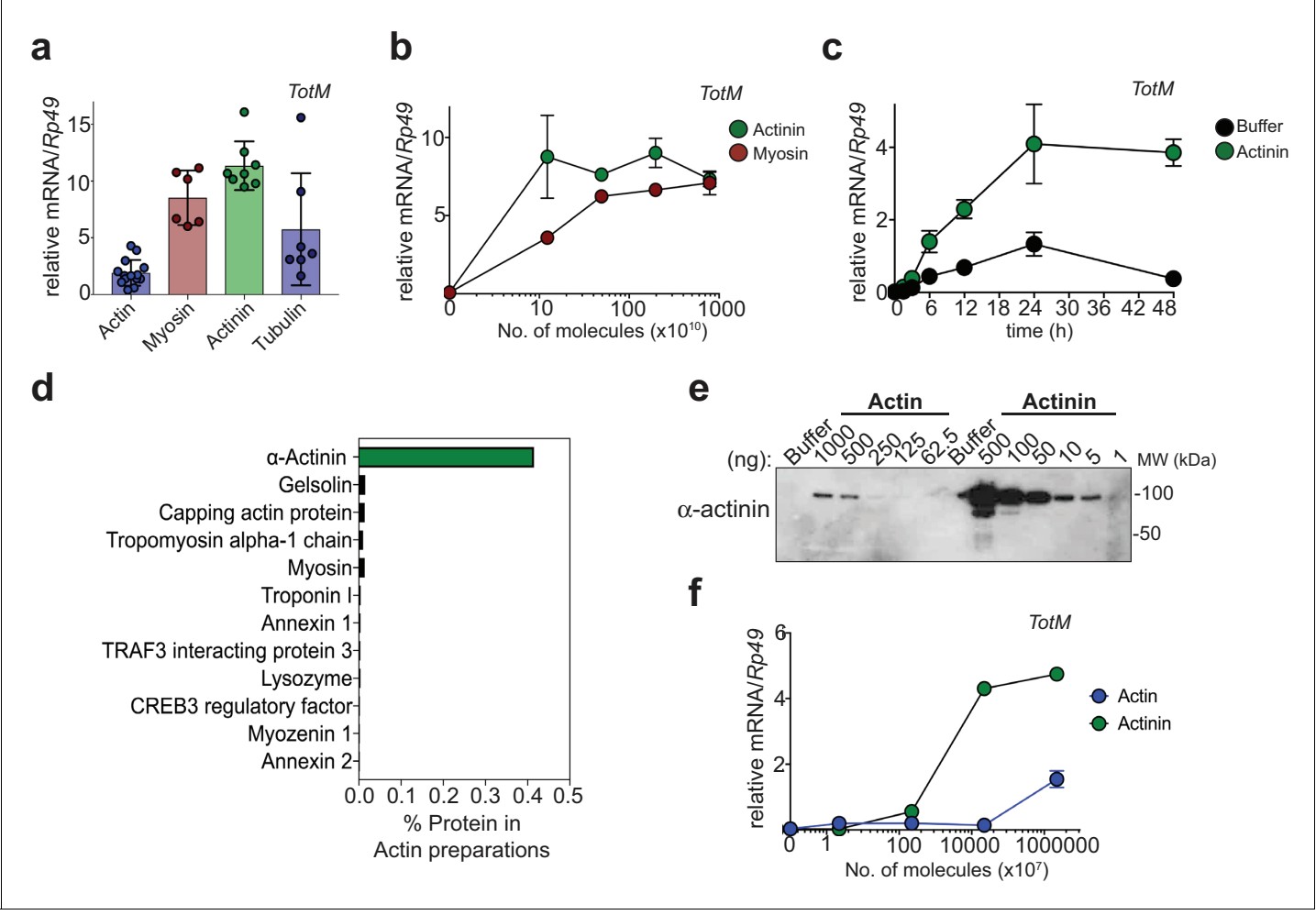

**Figure 1.** α-actinin is the most potent inducer of STAT-dependent genes. (**a**) $w^{1118}$ flies were injected with equimolar preparations of purified actin, α-actinin, myosin and tubulin ($9.4 \times 10^{17}$ molecules). Relative expression of *TotM* 24 hr post injection is shown. Data are pooled from two independent experiments with 10 flies/sample with at least triplicate samples. (**b**) $w^{1118}$ flies were injected with preparations containing the indicated number of molecules of purified myosin or purified α-actinin. Relative expression of *TotM* 24 hr post injection is shown. Data are representative of two independent experiments with 10 flies/sample with duplicate samples. (**c**) $w^{1118}$ flies were injected with buffer or 3.68 ng of purified α-actinin. Relative expression of *TotM* over a 48 hr period is shown. Data are representative of two independent experiments with 10 flies/sample with triplicate samples. (**d**) Purified actin was subjected to mass spectrometry analysis and contaminating proteins are expressed as % of protein preparation. (**e**) Indicated protein amounts (ng) of purified actin and α-actinin were analysed by western blot using an anti-α-actinin antibody. Data are representative of three independent experiments. (**f**) $w^{1118}$ flies were injected with preparations containing the indicated number of molecules of rabbit muscle purified actin or purified α-actinin. Relative expression of *TotM* 24 hr post injection is shown. Data are representative of three independent experiments with 10 flies/sample with duplicate samples. *TotM* relative levels were calculated using the housekeeping gene *Rp49* as a reference gene. Bars represent mean ± SD.

DOI: https://doi.org/10.7554/eLife.38636.002

responsive genes in *Drosophila*. Strikingly, injection of bacterially-expressed actin did not elicit the expression of *TotM* or of another STAT target gene, *Diedel*, unlike actin purified from rabbit muscle (**Figure 2c**). In contrast, bacterially-expressed α-actinin, like α-actinin purified from rabbit muscle, elicited the expression of various STAT-responsive genes, including *TotM*, *TotA* and *Diedel*, in a dose-dependent manner (**Figure 2d** and data not shown). Neither recombinant bacterially-expressed nor rabbit musclederived α-actinin elicited the expression of *Drs* and *Dpt*, Toll and Imd target genes, respectively, 24 hr post injection (**Figure 2—figure supplement 1a**), indicating that the proteins were devoid of microbial contaminants and confirming specificity for STAT targets. Lack of expression of *Drs* and *Dpt* by α-actinin was confirmed at multiple timepoints within a 48 hr period

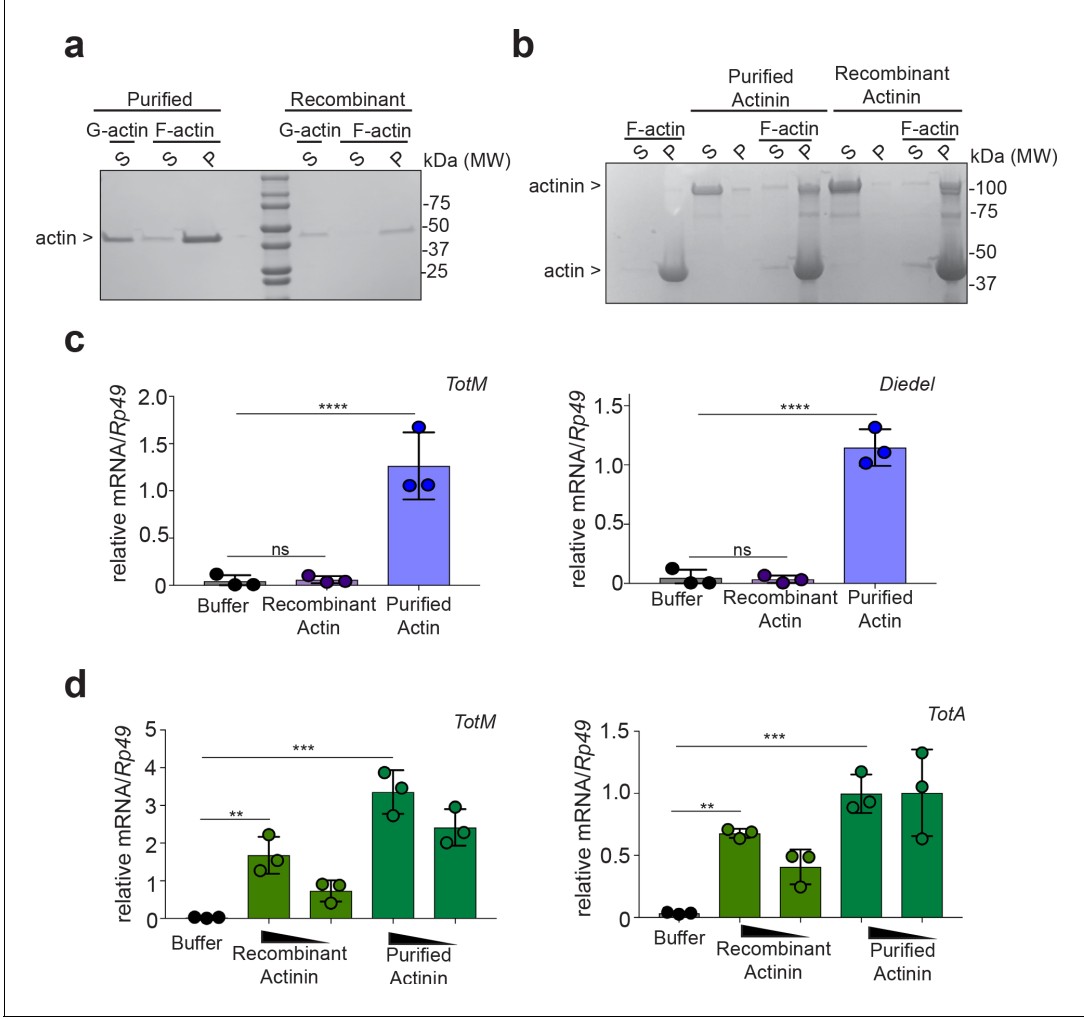

**Figure 2.** Bacterially-expressed recombinant α-actinin but not actin induces STAT-responsive genes. (a) Actin purified from rabbit muscle or recombinantly made in bacteria was incubated in either G-actin or F-actin buffers before subjecting samples to ultracentrifugation to separate globular and filamentous actin. Supernatant (S) and pelleted (P) fractions were analysed by Tris-Gylcine gels and visualized by Coomassie staining. Data are representative of two independent experiments. (b) α-actinin from rabbit muscle or recombinantly made in bacteria was added or not to polymerised F-actin. Then, samples were subjected to ultracentrifugation, and proteins in the supernatant (S) or pellet (P) were analysed by Tris-Gylcine gels and visualized by cCoomassie staining. (c) $w^{1118}$ flies were injected with PBS buffer or equal amounts of bacterially-expressed recombinant or rabbit muscle purified actin (11.04 ng per fly). Relative expression of *TotM* and *Diedel* 24 hr post injection is shown. Data are representative of two independent experiments with 10 flies/sample with triplicate samples. (d) $w^{1118}$ flies were injected with PBS buffer, or equal amounts of bacterially-expressed recombinant or rabbit muscle purified α-actinin (3680 pg or 368 pg per fly). Relative expression of *TotM* and *TotA* 24 hr post injection is shown. Data are representative of two independent experiments with 10 flies/sample with triplicate samples. *TotM* relative levels were calculated using the housekeeping gene *Rp49* as a reference gene. Bars represent mean ± SD. Statistical analysis was performed using one-way ANOVA with Sidak's multiple comparison test as post-test for pairwise comparisons. Significant differences with Sidak's multiple comparison test are shown (ns, not significant; *p<0.05; **p<0.01; ***p<0.001; ****p<0.0001).
DOI: https://doi.org/10.7554/eLife.38636.003

The following figure supplement is available for figure 2:

**Figure supplement 1.** Bacterially-expressed recombinant or rabbit muscle purified α-actinin do not induce the expression of antimicrobial genes.
DOI: https://doi.org/10.7554/eLife.38636.004

(*Figure 2—figure supplement 1b*). In contrast, septic injury (*Escherichia coli* and *Micrococcus luteus*) induced both *Drs* and *Dpt* by 24 hr (*Figure 2—figure supplement 1c*). Finally, RNAi-mediated knockdown of *Nox*, *Src42A* and *Shark* using three fat body-restricted drivers (c564, r4 and *Lpp*) abrogated the induction of STAT-responsive genes in response to injection of α-actinin (*Figure 3a*), whereas knockdown of *Nox*, *Src42A* and *Shark* using a haemocyte-restricted driver (HmlΔ) did not

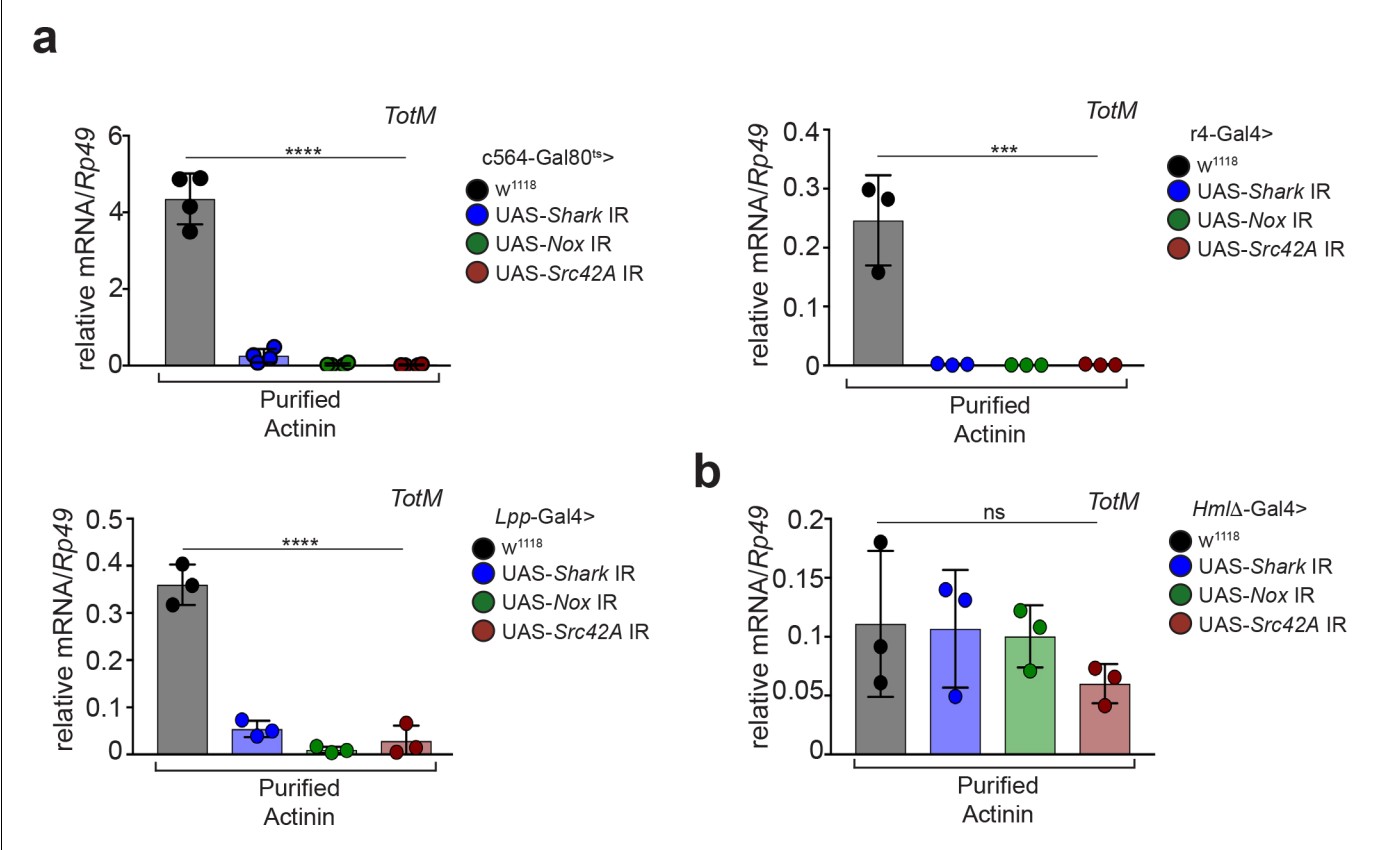

**Figure 3.** α-actinin induced STAT-dependent genes require *Nox*, *Shark* and *Src42A* expression in the fat body. (**a**) Flies in which *Shark*, *Nox*, or *Src42A* were knocked down in the fat body using three different driver lines (c564-Gal80ts, r4-Gal4 or *Lpp*-Gal4), or control flies lacking UAS-target sequences, were injected with either PBS buffer or rabbit muscle purified α-actinin (13.8 pg per fly). Relative expression of *TotM* 24 hr post injection is shown. Data are representative of two independent experiments with 5 – 10 flies/sample with at least triplicate samples. (**b**) Flies in which *Shark*, *Nox*, or *Src42A* were knocked down in haemocytes (*Hml*Δ-Gal4), or control flies lacking UAS-target sequences, were injected with either PBS buffer or rabbit muscle purified α-actinin (13.8 pg per fly). Relative expression of *TotM* 24 hr post injection is shown. Data are representative of two independent experiments with 5 – 10 flies/sample with triplicate samples. *TotM* relative levels were calculated using the housekeeping gene *Rp49* as a reference gene. Bars represent mean ± SD. Statistical analysis was performed using one-way ANOVA with Sidak's multiple comparison test as post-test for pairwise comparisons. Significant differences with Sidak's multiple comparison test are shown (ns, not significant; *p<0.05; **p<0.01; ***p<0.001; ****p<0.0001).
DOI: https://doi.org/10.7554/eLife.38636.005

alter the response (*Figure 3b*). Therefore, injection of bacterially-expressed or rabbit muscle α-actinin into *Drosophila melanogaster* is sufficient to induce the same response that we previously described to be elicited by injection of actin preparations. Taken together, these data suggest that α-actinin is a major bioactive component in cytoskeletal protein preparations, including actin.

We have previously reported that actin injection, mimicking extracellular cytoskeletal exposure, can trigger the expression of STAT target genes in *Drosophila* (*Srinivasan et al., 2016*). Actin is difficult to produce as a recombinant protein in bacteria and commercially-available actin (>99% pure) is traditionally purified from rabbit muscle or human platelets. Here, we demonstrate that the activity of such actin preparations is likely attributable to trace amounts of co-purifying α-actinin. This conclusion is supported by several findings: (1) α-actinin engages the same fat body Nox/Src42A/Shark pathway previously described for injected actin; (2) contamination with α-actinin can account stoichiometrically for the activity of actin preparations, α-actinin being more than one-hundred fold more potent that actin; (3) recombinant α-actinin expressed in bacteria retains activity while activity is lost for bacterially-expressed actin. These new findings support our earlier conclusion that exposure of cytoskeletal components is an evolutionarily-conserved sign of cell damage but suggest that the nature of the cytoskeletal element acting as a DAMP can vary across species. Thus, while (F-)

actin is sensed by at least one mammalian receptor, DNGR-1, in *Drosophila melanogaster* α-actinin appears to act as the dominant trigger for the STAT-dependent response we have studied. Consistent with the notion that α-actinin may be the common denominator of the response to extracellular proteins in *Drosophila*, preliminary mass spectrometry analysis suggests that trace amounts of α-actinin are also found in tubulin and myosin preparations (data not shown). The inability to deplete α-actinin using antibodies makes it difficult to address the issue of contamination, especially in the case of large multi-subunit proteins such as myosin for which recombinant expression in bacteria is not feasible. More work is needed to clarify whether any cytoskeletal proteins besides α-actinin act as inducers of STAT-responsive genes in *Drosophila* and to identify the putative receptor(s) responsible for initiating the response.

## Materials and methods

### Fly maintenance and injections

Fly maintenance, breeding, transgene induction, injections, septic injury, RNA extraction, cDNA synthesis and quantitative real-time PCR were performed as previously described (*Srinivasan et al., 2016*).

The following stocks were used:

| Fly stock | Description |
|---|---|
| $w^{1118}$ | Control strain |
| ;UAS-*Shark* IR; (Shark Fr [RNAi]) | Interfering RNA for knockdown of *Shark*. Kindly donated by Marc Freeman. |
| w;c564-Gal4;*Tub*-Gal80[ts] | Temperature-sensitive fat body-specific driver line. |
| ;UAS-*Src42A* IR | VDRC ID: 100708 |
| ;UAS-*Nox* IR | VDRC ID: 100753 |
| ;;r4-Gal4/TM6C.Sb[1] | Fat body-specific driver line |
| ;;*Lpp*-Gal4/TM6.Sb[1] | Fat body-specific driver line |
| *Hml*Δ-Gal4/CyO | Haemocyte-specific driver |

### Purified cytoskeletal proteins

Purified cytoskeletal proteins were reconstituted and stored as advised by the supplier (Cytoskeleton Inc).

### Bacterial expression of cytoskeletal proteins

Rosetta-2 BL21 *E. coli* bacteria were transformed with pET8c-His$_6$-human Actinin-2 as described (*Ribeiro et al., 2014*). A starter culture was made by inoculating Ampicillin-containing LB with a well-defined colony and incubating overnight at 37°C. Large-scale expression was carried out using Overnight Express LB medium (Novagen) supplemented with 10 ml glycerol inoculated with 2 ml of the starter culture. Cell growth and protein expression was carried out at 30°C for 16 hr. Cells from these high-density shaking cultures ($OD_{600}$ > 8 A.U) were harvested by centrifugation for 15 min at 7500 x *g*. For purification of the His-tagged protein, the bacterial pellet was re-suspended in 50 ml of lysis buffer containing 50 mM Tris-HCl (pH 8.0), 150 mM NaCl, 0.1% Triton X-100 and antiproteases (Roche). Cells were disrupted by sonication (5 × 30 s) and lysates were centrifuged at 18,000 x *g* for 30 min at 4°C. Clarified lysates were incubated with 3 ml Ni-NTA agarose at 4°C for 16 hr on rotation. Beads were washed three times for 5 min at 4 °C with 25 ml wash buffer (50 mM Tris-HCl (pH 7.5), 150 mM NaCl, antiproteases, 5 mM Imidazole) . After the final wash, beads were incubated with 10 ml buffer A to elute His-tagged proteins (50 mM Tris-HCl (pH 7.5), 150 mM NaCl, antiproteases, 2 mM beta-mercaptoethanol, 250 – 500 mM Imidazole). Two elutions were collected and subsequently purified to homogeneity by anion exchange and size exclusion chromatography. To that end, α-actinin eluted from the Ni-NTA affinity resin was dialysed into buffer containing 50 mM Tris-HCl (pH 7.5), 2 mM beta-mercaptoethanol and applied to a Mono Q 5/50 anion exchange

column (GE Life Sciences) equilibrated with 50 mM Tris (pH 7.5) and 1 mM DTT. Bound α-actinin was eluted by application of a NaCl gradient from 0 to 1M over 30 column volumes. The peak containing α-actinin was determined by SDS-PAGE, the relevant fractions pooled and concentrated to 0.5 ml and applied to a S75 10/300 size exclusion column equilibrated with PBS (pH 7.4) and 1 mM DTT. Fractions containing α-actinin were pooled and stored until required. Recombinant bacterially-expressed human smooth muscle actin was purchased from a commercial source (ab134555, Abcam).

## Mass spectrometry

Protein preparations were subjected to SDS/PAGE and migrated approximately 2 cm into the gel. The gel lanes were excised and proteins were in-gel digested using trypsin. Tryptic peptides were analysed with an LTQ Orbitrap-Velos mass spectrometer coupled to an Ultimate3000 HPLC equipped with an EASY-Spray nanosource (Thermo Fisher Scientific). Raw data was processed using MaxQuant v1.3.05 with intensity based absolute quantification (iBAQ) selected as the quantification algorithm. The output table was imported into Perseus software v1.4.0.2, and percentages of contamination were calculated using the untransformed iBAQ values.

## Actin pelleting assay

Performed as described by the supplier (Cytoskeleton). Briefly, actin was diluted in modified G-actin buffer (5 mM Tris HCl pH 8.0, 0.2 mM $CaCl_2$, 0.2 mM ATP, 0.5 mM DTT) and incubated on ice for 60 min for complete depolymerisation of actin oligomers. Subsequently, samples were supplemented with F-actin buffer (10 mM Tris-HCl pH 7.5, 50 mM KCl, 2 mM $MgCl_2$ and 1 mM ATP) incubated for 60 min at RT. Samples were then centrifuged for 1 hr at 100,000 x $g$, and protein levels in the supernatant and pellet were assessed by SDS-PAGE and Coomassie staining. For α-actinin pelleting assays the same protocol was used, with the only modification that α-actinin was added to the polymerising actin filaments.

## Western blotting

Protein samples were resuspended in 1X Laemmli SDS-PAGE buffer and boiled for 7 min at 95˚C. Protein was resolved on Tris-Glycine 4 – 20% precast gels (Bio-Rad). α-actinin and actin was detected via immunoblotting with anti-actin (Clone C4, MAB1501, Millipore) and anti-actinin (3134S, Cell Signaling Technology) antibodies.

## Acknowledgements

We thank Nic Tapon, Paul Martin, Maxine Holder, Georgina Fletcher, Ieva Gailite and members of the Immunobiology Laboratory for helpful discussions and suggestions. We are grateful to Kristina Djinovic-Carugo for the α-actinin plasmid. We thank the Francis Crick Institute Genomics Equipment Park, Proteomics, Structural Biology, and the Fly facility for assistance. We also thank the Vienna Drosophila Resource Center for *Drosophila* lines and Flybase for online resources. This work was supported by The Francis Crick Institute, which receives core funding from Cancer Research UK (FC001136), the UK Medical Research Council (FC001136), and the Wellcome Trust (FC001136), and by Investigator Award WT106973MA from the Wellcome Trust to CRS. LT is funded by the Fundação para a Ciência e Tecnologia (www.fct.pt) grant PTDC/BEX- GMG/3128/2014. CMH was supported by a long-term fellowship from the Federation of European Biochemical Societies (FEBS).

## Additional information

### Funding

| Funder | Grant reference number | Author |
|---|---|---|
| Francis Crick Institute | FC001136 | Caetano Reis e Sousa |
| Wellcome Trust | WT106973MA | Caetano Reis e Sousa |
| Federation of European Biochemical Societies | | Conor M Henry |

| Fundação para a Ciência e Tecnologia | PTDC/BEX- GMG/3128/2014 | Luis Teixeira |

The funders had no role in study design, data collection and interpretation, or the decision to submit the work for publication.

## Author contributions
Oliver Gordon, Conor M Henry, Conceptualization, Investigation, Visualization, Writing—original draft; Naren Srinivasan, Conceptualization, Writing—review and editing; Susan Ahrens, Anna Franz, Safia Deddouche, Probir Chakravarty, David Phillips, Luis Teixeira, Barry Thompson, Marc S Dionne, Will Wood, Involved in the early phases of the project; Roger George, Svend Kjaer, Resources; David Frith, Ambrosius P Snijders, Investigation; Rita S Valente, Carolina J Simoes da Silva, Validation; Caetano Reis e Sousa, Conceptualization, Supervision, Funding acquisition, Writing—original draft, Writing—review and editing

## Author ORCIDs
Oliver Gordon (ID) http://orcid.org/0000-0002-2567-2482
Conor M Henry (ID) http://orcid.org/0000-0001-7504-5121
Luis Teixeira (ID) https://orcid.org/0000-0001-8326-6645
Barry Thompson (ID) https://orcid.org/0000-0002-0103-040X
Marc S Dionne (ID) https://orcid.org/0000-0002-8283-1750
Caetano Reis e Sousa (ID) http://orcid.org/0000-0001-7392-2119

## Decision letter and Author response
Decision letter https://doi.org/10.7554/eLife.38636.011
Author response https://doi.org/10.7554/eLife.38636.012

---

## Additional files
### Supplementary files
• Transparent reporting form
DOI: https://doi.org/10.7554/eLife.38636.007

### Data availability
Data generated or analysed during this study are included in the manuscript. Mass spectrometry data were uploaded as supporting file.

---

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
