## [Decision Letter]

Thank you for submitting your article "The cytoskeletal protein α-actinin accounts for the bioactivity of actin preparations in inducing STAT target genes in *Drosophila melanogaster*" for consideration by *eLife*. Your article has been reviewed by three peer reviewers, including Bruno Lemaître as the Reviewing Editor and Reviewer #1, and the evaluation has been overseen by Tadatsugu Taniguchi as the Senior Editor.

The reviewers have discussed the reviews with one another and the Reviewing Editor has drafted this decision to help you prepare a revised submission.

Summary:

This study is a follow up to the authors recent work, which demonstrated that actin is a DAMP in *Drosophila*. This finding is expanded on here, with the interesting observation that actin is not actually responsible for the immunostimulatory activity observed. Rather, an actin binding protein, α-actinin, is responsible. The best evidence supporting this conclusion comes from the new studies presented, whereby actin that was purified from rabbit muscle contained immunostimulatory activity, but actin purified from bacteria did not. This finding, coupled to studies demonstrating that pure preparations of α-actinin are more stimulatory than "contaminated" actin preps from rabbit, are compelling. Genetic evidenced is presented to demonstrate that the pathway activated by α-actinin is the same as that described previously for actin.

Essential revisions:

1) It is fortunate that the same authors can identify themselves that this was not actin but α-actinin the elicitor. Their conclusion would be reinforced if another laboratory could reproduce independently their observations. Was the injection experiment carried out independently in the Wood, Dionne or Teixeira lab (order the product and monitor *upd3* upon injection)? This should be the case and this should be mentioned in the paper.

2) Many studies have shown that *upd3* is produced by hemocytes not the fat-body (Agaisse, 2003; Chakrabarti, 2016). C564 is a Gal4 driver that target both hemocyte and fat body. I would recommend that the authors test their hypothesis using more specific drivers: *hml*Gal4 (hemocyte) and *Lpp*-Gal4 (fat body) to reinforce their conclusion that fat body not hemocytes is sensing actinin.

3) The three reviewers consider that the present paper is more a correction (of a mistake) than an extension. This should be mentioned in the Abstract and text (Introduction, 'corrected' instead of 'extend').

---

## [Author Response]

Essential revisions:1) It is fortunate that the same authors can identify themselves that this was not actin but α-actinin the elicitor. Their conclusion would be reinforced if another laboratory could reproduce independently their observations. Was the injection experiment carried out independently in the Wood, Dionne or Teixeira lab (order the product and monitor upd3 upon injection)? This should be the case and this should be mentioned in the paper.

As requested, we asked for the experiment to be independently reproduced by the Dionne and Teixeira lab. Both labs carried out experiments as per our protocol but using α-actinin that they purchased from Cytoskeleton (independently from each other and independently from the Reis e Sousa lab). Both the Dionne and Teixeira labs reproduced the observation that intrathoracic injection of α-actinin results in upregulation of STAT-responsive genes such as *TotM* (assessed by both labs) and *TotA* (additionally assessed by the Teixeira lab). A sentence was added to the manuscript mentioning these experiments (Results and Discussion, first paragraph). The results of one of two independent experiments per lab are shown below in Author response image 1 and Author response image 2 (figure legends provided by the originators).

**Author response image 1. respfig1:** Injection experiment performed in the Dionne lab. Relative expression of *TotM* 18hr post injection. Data are representative of two independent experiments with 3 flies/sample. Relative levels of expression were calculated using the housekeeping gene *Rlp1* as a reference gene. Bars represent mean ± SD.

**Author response image 2. respfig2:** Injection experiment performed in the Teixeira lab. 3-6 day old *w1118* DrosDel isogenic (Ryder et al., 2004; Chrostek et al., 2013) females were injected with 36.8nL of either buffer (4mM Tris-HCL pH 7.6, 4mM NaCl, 20µM EDTA, 1% (w/v) sucrose and 2% w/v dextran) or α–Actinin in buffer (1μg/μl, Cytoskeleton, #027AT01-A). Flies were collected 24h post infection and RNA extracted from 5 pools of 10 flies per condition, using tripleXtractor reagent (GRiSP, # GB23.0200) followed by DNase treatment (Promega, #M6101). cDNA synthesis was performed with M-MLV Reverse Transcriptase (Promega, #M1705) and Random Primers (Promega, #C1181). cDNA was diluted ten times in DEPC water (Invitrogen, #46-2224) and analysed for gene expression by qPCR using iTaq Universal SYBR Green Supermix (Bio-rad, #1725125). Reactions were carried out using a QuantStudio 7 Flex machine. Relative expression ratios of *TotM* (**A**) and *TotA* (**B**) were calculated with the Pfaffl method (Pfaffl, 2001), using *Rp49* as reference gene and buffer injected flies as control values. *TotM* and *TotA* expression is induced by Actinin (linear mixed model, *p* < 0.001).

2) Many studies have shown that upd3 is produced by hemocytes not the fat-body (Agaisse, 2003; Chakrabarti, 2016). C564 is a Gal4 driver that target both hemocyte and fat body. I would recommend that the authors test their hypothesis using more specific drivers: hmlGal4 (hemocyte) and Lpp-Gal4 (fat body) to reinforce their conclusion that fat body not hemocytes is sensing actinin.

As suggested by the reviewers, we have repeated knockdown of *Nox*, *Src42A* or *Shark* using additional fat body drivers (*Lpp*-Gal4 and r4-Gal4), as well as a haemocyte driver (*Hml*Δ-Gal4). Knockdown with all three fat body drivers (c564-Gal4, *Lpp*-Gal4 and r4-Gal4) resulted in complete or nearly complete loss of *TotM* induction upon injection of α-actinin, whereas knockdown using *Hml*Δ-Gal4 did not have an appreciable effect. These data are now included in Figure 3 of the manuscript. We tried to measure *upd3* induction by RT-qPCR, but did not obtain clear cut results – we discussed our inability to convincingly measure *upd3* transcripts in our previous study (Srinivasan et al., 2016). However, in our previous study we knocked down *upd3* specifically in haemocytes (or depleted haemocytes altogether) and found that this had no impact on induction of STAT-responsive genes (Srinivasan et al., 2016 – Figures 5C, D). Although those experiments utilised purified actin and not α-actinin, we believe that those data, together with the experiments included in the present study, support the notion that haemocytes are dispensable for the response to cytoskeletal protein injection, which involves the fat body.

3) The three reviewers consider that the present paper is more a correction (of a mistake) than an extension. This should be mentioned in the Abstract and text (Introduction, 'corrected' instead of 'extend').

We are uncomfortable with stating that we are “correcting a mistake” as it can be read to mean that either the data in the previous paper were incorrect or that we made an actual mistake (such as misread actin for actinin on a tube label). In actual fact, we are revising the interpretation of those data and refining the conclusions. We therefore have altered the Abstract and Introduction, as requested, to state that we revise our earlier conclusion/revise interpretation of the earlier data.